# Unraveling the influence of defects on Sulfonamide adsorption onto Blue-phosphorene nanotube using density functional theory

José M. Vergara[1◉], Julian D. Correa[1◉*], Miguel E. Mora-Ramos[2◉], Elizabeth Flórez[1◉]

**1** Facultad de Ciencias Básicas, Universidad de Medellín, Medellín, Colombia, **2** Centro de Investigación en Ciencias, Instituto de Investigación en Ciencias Básicas y Aplicadas, Universidad Autónoma del Estado de Morelos, Cuernavaca, Morelos, México

◉ These authors contributed equally to this work.

* jcorrea@udemedellin.edu.co

**Data availability statement:** All relevant data are within the manuscript and its Supporting information files.

**Funding:** This research was supported by the Ministerio de Ciencia, Tecnología e Innovación (MINCIENCIAS) GRAND 120680864729 to JC.

**Competing interests:** The authors have declared that no competing interests exist.

## Abstract

Sulfonamide antibiotics are commonly used in human therapy. Consequently, pharmaceutical residues may seep into the surface and groundwater, contaminating the aquatic environment. Adsorption is the most widely used method for removing these contaminants from water bodies. This study investigates the efficiency of (14, 14) armchair and (14, 0) zigzag blue phosphorene-based nanotubes (BPNT) as adsorbents of three popular toxic antibiotics, Sulfanilamide (SAM), Sulfadimethoxine (SMX), and Sulfadiazine (SDZ), from water bodies. All calculations are performed using density functional theory. Analyzed molecules are weakly adsorbed on the pristine BPNTs with an adsorption energy of about –0.312, –0.285, and –0.377 eV. Further, electronic properties of the antibiotics-adsorbed BPNTs are investigated. The effect of single-vacancy BPNTs on the adsorption affinity of antibiotic molecules was studied. Compared with pristine systems, despite the increase in reactivity of zigzag BPNTs to the sulfonamides, armchair configurations show a transition from bipolar-magnetic semiconductor to a non-magnetic metallic system, suggesting that defective armchair BPNTs can also be employed as a sensor for antibiotic molecules. Single-vacancies increase the $E_{ads}$ values of all studied systems by up to 89%, indicating an improvement in the capacity of BPNTs to adsorb these biologically active sulfonamide-based compounds.

## Introduction

Sulfonamide molecules are an important class of drugs, including various types of pharmacological agents. They possess antibacterial, antiviral, anticarbonic anhydrase (CA), diuretic, protease inhibitors, cyclooxygenase 2 (COX2) inhibitors, anticancer activities, and others [1]. These antibiotics are produced in large quantities and are widely used in human therapy and livestock production. As a result, increasing amounts of pharmaceutical residues from sewage

effluents, hospital effluents, and other untreated wastewater will seep into surface and groundwater, then migrate and transform in the aquatic environment. The presence of this type of residual substance in aqueous environments generates great concern due to its potential risk to the environment and biological survival because it represents a high risk to human health and can affect the evolutionary structure of the bacterial community, strengthening bacterial resistance to these products [2–4].

Numerous methods are utilized to remove antibiotics from water, including biological processes, advanced oxidation processes (AOPs), and adsorption processes. Adsorption is considered one of the most advantageous techniques due to its simplicity, economy, and easy operation. For antibiotic remediation, a wide range of adsorbents have been used [2,3]. Usually, they involve activated carbon and carbonaceous materials. Notably, carbon-based materials, such as biochar, graphene, and nanotube carbon have emerged as the most promising solutions for removing antibiotics from contaminated waters. Active carbon is characterized by its functionality, porosity, surface morphology, and chemistry, which enhance the adsorption process. [5]. On the other hand, two-dimensional carbon materials such as graphene or graphene oxides show an affinity for removing various classes of organic contaminants from water due to their highly hydrophobic surface, open-layer morphology, and high adsorption affinity [6]. It has been discovered that carbon nanotubes have the potential to be used in removing antibiotics from water. This is due to their cost-effectiveness, lower energy requirements, minimal chemical usage, and environmental impact, as well as their large surface area and increased chemical reactivity [7,8]. The adsorption of Sulfanilamide (SAM), Sulfamerazine (SMR), Sulfadimethoxine (SMX), Sulfadiazine (SDZ), Sulfamethazine (SMT), and Sulfamethoxydiazine (SMD) on carbon nanotubes has been explored in several works [9–11]. In the case of SMX, it is shown that the $\pi$ - $\pi$ interaction was one of the mechanisms for SMX adsorption on multi-wall carbon nanotubes, and that, in general, the antibiotics are adsorbed in CMs through various non-covalent interactions, including van der Waals dispersion, $\pi$ - $\pi$ interactions, hydrophobic interaction, and hydrogen bonding [9]. Theoretical calculations based on density functional theory show that SDZ and SMX sulfonamides are adsorbed on single wall carbon nanotube with adsorption energies of –0.566 eV and –0.551 eV, respectively, but when a water environment is considered these energies increase [11]. More recently, Liu et al. have shown that carbon nanotubes have the potential to efficiently remove sulfonamides from aqueous solutions through the adsorption process, reaching a high efficiency (in a pH adsorption range of 3 to 9). These studies show that carbon nanotubes have a promising potential as an effective adsorbent for removing sulfonamide antibiotics from aqueous solutions. However, these processes require further investigation, with the inclusion of novel materials, to guide engineering applications since removing antibiotics can be, in some cases, incomplete [12,13].

Taking into account the benefits of carbon nanotubes, the performance of blue phosphorene (BP) nanotubes in antibiotic adsorption processes has been investigated in recent years. BP is one of the four main 2D allotropes of phosphorus (black, red, green are the other three), predicted to be thermal and dynamically stable back in 2014 [14]. It has already been experimentally realized [15–17]. Regarding blue-phosphorene nanotubes (BPNTs), Bhuvaneswari et al. [18], studied the molecular interaction of (10, 10) armchair-type blue phosphorene nanotubes with Sulfapyridine (SP) which is an antibacterial drug, found that the interaction properties of the complex such as remarkable Bader charge transfer, significant adsorption energy, and the prominent average band gap variation, indicate a high efficiency for the use of BPNTs to eliminate toxic antibiotics from water bodies.

Generally, to activate the surface of nanotubes, whether made of carbon or other material, it is necessary to either induce defects, dope, decorate, or functionalize the nanotube wall to

achieve an increase in its adsorption capacity [19–23]. In the case of structural defects, they could be induced by irradiating the structure with ions or electrons [24]. Such defects produce modifications in the electronic properties of the system. For BP monolayers, it was shown that the presence of structural defects favors the adsorption of volatile organic molecules [25]. Hence, it would be interesting to analyze and compare the effect of defect-laden BPNTs on the adsorption capacities for molecules of interest.

Despite theoretical efforts, there are still untreated aspects regarding the performance of BPNTs in the removal of antibiotics. Therefore, in the present work, their adsorption behavior has been evaluated, considering the effect that inclusion of structural defects such as single-vacancies could have for this type of application. Specifically, we focus on the adsorption of sulfonamide molecule antibiotics.

## Computational details

We analyze the adsorption of three toxic antibiotics, Sulfanilamide (SAM), Sulfadimethoxine (SMX), and Sulfadiazine (SDZ)), on pristine and defective single vacancy (SV) BPNTs. The purpose is to contrast and identify the influence of SV in the adsorption capacity of the nanotube. Our analysis considers BPNTs of armchair (AM) and zigzag (ZZ) types, characterized by the chiral numbers (14, 14) and (14, 0), respectively. Distinct chiral numbers defined different electronic properties of the BPNTs.

The calculation of structural and electronic properties is performed within the framework of DFT, as it is implemented in the *ab initio* SIESTA package [26], where localized double-$\zeta$ polarized atomic orbitals are employed as basis set, together with conserved norm pseudopotentials. A supercell of $1 \times 1 \times 5$ guarantees that the molecules do not interact with their images. The analyzed adsorption sites can be seen in Fig 1. For the exchange and correlation functional we employ the one proposed by Dion et al. [27] whose exchange part was modified by Klimes et al. [28]. This functional include van der Waals interactions and is usually labeled as KBM. All structures are relaxed through the FIRE minimization algorithm, until the force on the atoms is smaller than 0.04 $eV$/Å [29]. The Brillouin zone sampling is carried out via Monkhorst-Pack-grid of $1 \times 1 \times 2$. A mesh cutoff value of 350 $Ry$ was set. The study is carried out with the inclusion of collinear spin-polarization for SV systems and without spin-polarization for pristine BPNT cases.

The adsorption energy between the molecule and the BPNT is characterized by evaluating the adsorption energy ($E_{ads}$), which is defined as:

$$E_{ads} = E_{Total} - E_{BPNT} - E_{Adsorbate},\qquad(1)$$

where $E_{Adsorbate}$ is the total energy of the antibiotic, $E_{Total}$ is the total energy of the BPNT+adsorbate complex and $E_{BPNT}$ is the BPNT total energy.

Recovery time captures the temporal cost for desorption of a target molecule from the sensing material's surface [30]. It can be determined according to the transition state theory and Van't Hoff–Arrhenius explanation. Adopting this approach to evaluate the recovery time of BPNTs from the interaction with toxic antibiotics [31,32] implies using the following equation, which encompasses the attempt frequency $\theta_0$, adsorption energy from Eq 1, Boltzmann's constant ($k_B$), and the temperature ($T$):

$$\tau = \theta_0^{-1} exp\left(\frac{-E_{ads}}{k_B T}\right),\qquad(2)$$

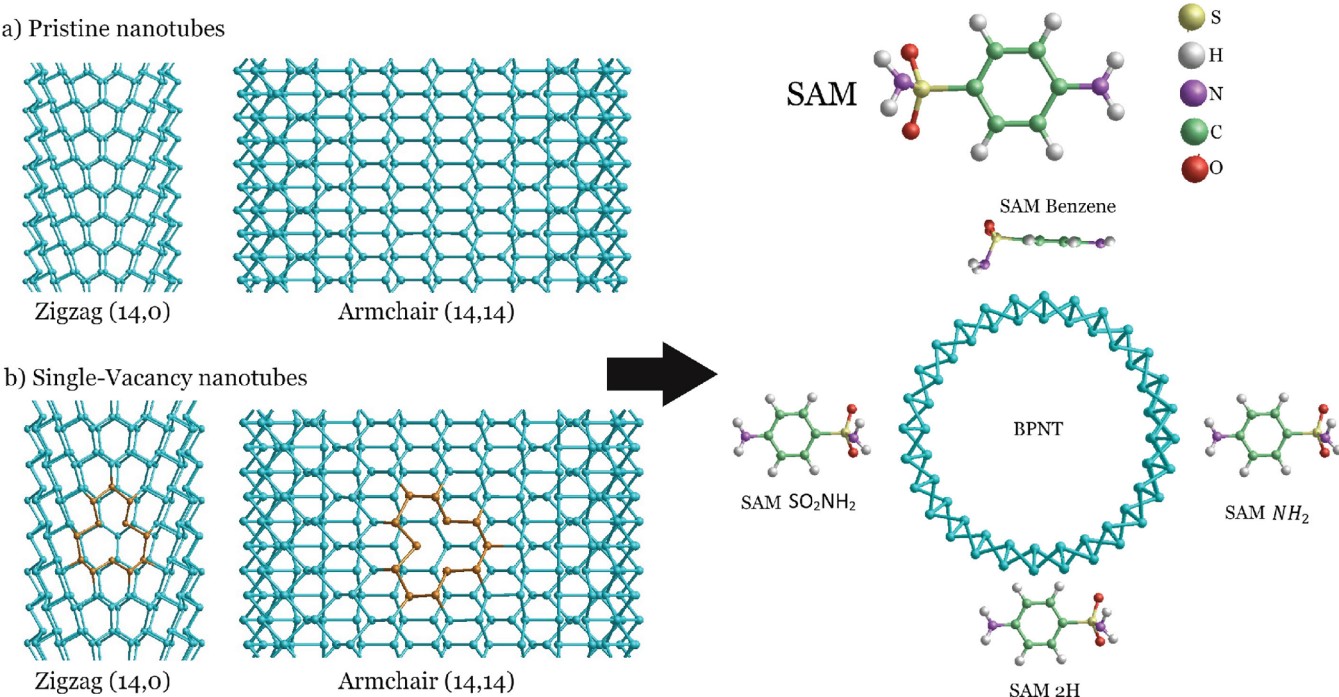

**Fig 1. Schematic view of all evaluated SAM adsorption sites for both, pristine and single-vacancy blue phosphorene nanotubes.**

https://doi.org/10.1371/journal.pcbi.0312034.g001

where the frequency factor $\theta_0$ is assume to be $10^{12} Hz$ and $10^{16} Hz$, under visible and UV light conditions, respectively [30,33,34], $k_B$ is approximately $8,617 \times 10^{-5} eV/K$ [30] and $T$ is taken to be equal to 300, 310, 320 and, 330 $K$.

To estimate the sensibility of the nanotube to a given molecule, we can employ the proportionality relation between conductivity and energy band gap.

$$\sigma \propto \exp{-E_{gap}/2KT}, \qquad (3)$$

being $E_{gap}$ the energy gap, $K$ the Boltzmann constant and $T$ the temperature. With this in mind, it is possible to define the sensitivity as

$$S = \frac{\sigma_{complex} - \sigma_{NT}}{\sigma_{NT}} = \exp{\Delta E_g/2KT} - 1 \qquad (4)$$

The effects of water on the adsorption energy of each antibiotic onto the nanotube surface are considered by employing the extended tight-binding(xTB) quantum chemistry method, implemented by Grimme's group [35]. In this implementation, it is possible to employ several parameterizations, covering a broad spectrum of elements of the periodic table. For our calculations, we employed the one denominated as GFN-0 [36], which is suitable to describe geometries, frequencies, and non-covalent interactions of large molecular systems. Such parameterization is also compatible with the implicit solvation model also implemented on xTB [37]. This combination of empirical models allows us to estimate the water medium's effect on the adsorption of sulfonamide-based compounds onto BPNTs.

## Results

### Geometric structures of Sulfonamide complexes on pristine and single-vacancy BPNTs

We report on the adsorption of SAM molecule on pristine and SV BPNTs for both armchair and zigzag configurations is. These BPNTs are obtained by the rolling of Blue-Phosphorene monolayer along one of its symmetric directions (zigzag or armchair), as is shown in the Supporting information S1 Fig. Adsorption has been analyzed considering four different target sites, indicated in Fig 1. As mentioned, the crucial property that supports verifying the kind of interaction taking place between antibiotics and the BPNT -thereby highlighting the use of designed base nanotube-, is the adsorption energy ($E_{ads}$) [38,39]. Results for $E_{ads}$ of the adsorption modes of SAM appear in Table 1, while for the rest of the molecules, we refer the reader to the Supporting information (S1 and S2 Tables). It is evidenced that, while SAM adsorption on $2H$, $NH_2$, and $SO_2NH_2$ sites is possible, Benzene position reveals as the most relevant from the outcome, since it entails the highest adsorption energy within its respective groups (same chirality and same type of vacancy); i.e. it is the most stable one in all evaluated systems. This is attributed to the presence of polar groups and aromatic rings offering $\pi - \pi$ interactions between target contaminants and BPNTs [3,4,40–42]. The polarity of the adsorption/interaction energy (negative) suggests that the process of molecule adsorption is stable and means an easier attraction of the SAM antibiotic towards both pristine and SV BPNTs. In the case of SV systems, there is an increment in $E_{ads}$ value compared with pristine the system for all studied positions. This indicates that structural defects, -in this case, single vacancies- slightly improve the ability of BPNTs for SAM molecule adsorption.

It can be noted that, for the SV system, ZZ chirality is most favorable than AM one, since the molecule adsorption energy is higher in ZZ BPNTs. That is, for each equivalent position of all evaluated systems, $E_{ads}$ has a percentage increase of almost 40% over the AM counterpart. This indicates that this particular geometry is, possibly, more active than the AM type and, therefore, it benefits the adsorption of SAM molecules.

Additionally, adsorption of SDZ and SMX antibiotics was studied on Benzene position as well, as can be seen in Fig 2. According to the obtained results, shown in Table 2, pristine BPNTs portrays a similar behavior in all evaluated molecules. That is, the AM BPNTs exhibit greater values of $E_{ads}$ than ZZ systems. This indicates that AM configurations should be a better option; and it is important to remark that AM phosphorene nanotubes present a significant stable nature in comparison to the ZZ phosphorene nanotubes [43].

It can be observed from Table 2 that the ordering of maximum adsorption capacity of the three sulfonamides is SAM > SDZ > SMX for pristine systems and ZZ SV. In SV AM-BPNTs

**Table 1. Results for the interaction of pristine and single-vacancy (SV) armchair and zigzag blue phosphorene nanotubes with SAM molecule; Adsorption energy ($E_{ads}$) is given in $eV$ units.**

| System | Initial SAM positions | Final SAM positions | $E_{ads}$ | System | Initial SAM positions | Final SAM positions | $E_{ads}$ |
|---|---|---|---|---|---|---|---|
| Pristine ZZ | **Benzene** | **Benzene** | **-0.279** | SV ZZ | **Benzene** | **Benzene** | **-0.509** |
| | $NH_2$ | $NH_2$ | -0.191 | | $2H$ | $2H$ | -0.497 |
| | $SO_2NH_2$ | $SO_2NH_2$ | -0.130 | | $NH_2$ | $NH_2$ | -0,380 |
| | $2H$ | $2H$ | -0.094 | | $SO_2NH_2$ | $SO_2NH_2$ | -0.272 |
| Pristine AM | **Benzene** | **Benzene** | **-0.312** | SV AM | **Benzene** | **Benzene** | **-0.373** |
| | $2H$ | $2H$ | -0.205 | | $2H$ | $2H$ | -0.221 |
| | $NH_2$ | $NH_2$ | -0.053 | | $NH_2$ | $NH_2$ | -0,161 |
| | $SO_2NH_2$ | $SO_2NH_2$ | -0,028 | | $SO_2NH_2$ | $SO_2NH_2$ | -0.122 |

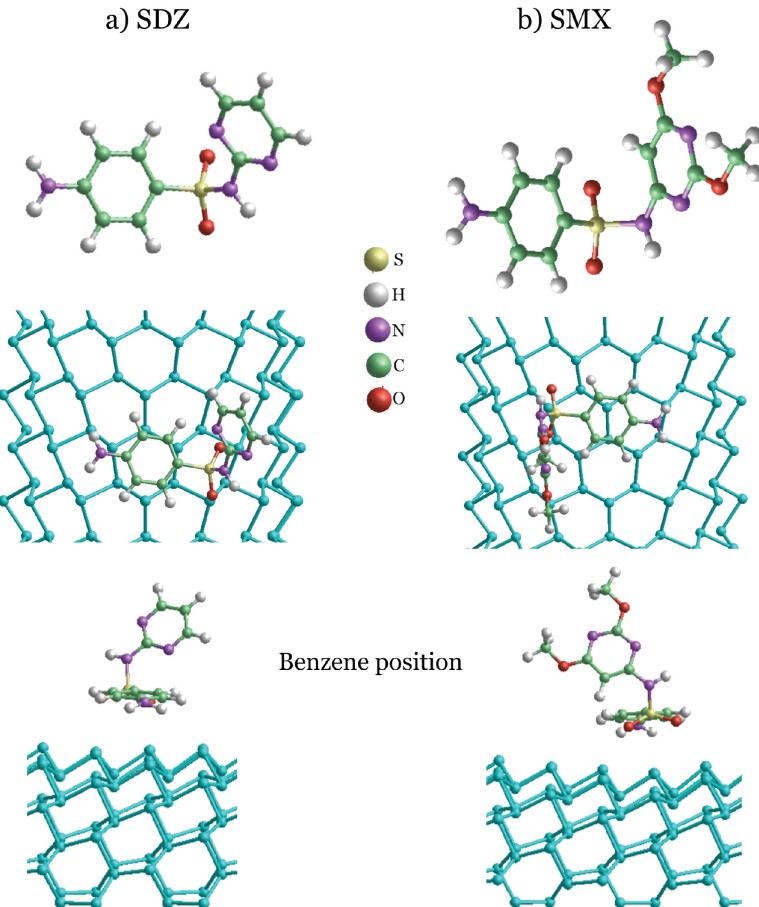

**Fig 2. Schematic view of evaluated Benzene position for SDZ and SMX adsorption sites for both, pristine and single-vacancy blue phosphorene nanotubes.**

https://doi.org/10.1371/journal.pcbi.0312034.g002

SDZ molecule presents higher $E_{ads}$ than SAM and SMX (SDZ > SAM > SMX). In comparison with results obtained in previous studies for BPTNs [39], SAM, SDZ, and SMX molecules have a smaller $E_{ads}$ than another antibiotics like oxytetracycline (OC) and sulfapyridine (SP) (< −1, 25 approximately). Regarding CNTs, previously reported absorption affinities depends on initial conditions, so that the order of adsorption efficiency of some sulfonamides was SMD > SMT > SDZ > SMX > SMR > SAM, as obtained by Liu et al. [10], SMT > SMR > SP > SDZ, as reported byby Zhao et al. [4]; and SMX > SMT obtained by Wei et al. [44]. Furthermore, according to the values obtained here, single vacancies improve adsorption energy of the SAM, SDZ, and SMX molecules, which means that the increase in $E_{ads}$ found in percentage ranges that go from 20% to 89% for the AM SAM (smaller increase) and AM SDZ (larger increase) systems. This indicates that structural defects, in this case, single vacancies, improve the capacity of BPNTs for adsorption of these toxic antibiotics.

## Electronic properties

The electronic properties of the system can be inspected through band structure. Here, we present the calculated band diagrams for clean and molecule-adsorbed (at the most stable

**Table 2. Results for the interaction of Benzene sites for pristine and single-vacancy (SV) with SAM, SDZ and SMX molecules, including calculated Bader charges ($\Delta_Q$). Adsorption energy ($E_{ads}$) is presented in *eV* units. The CIF file corresponding to each complex is in the Supporting information 2 CIF Files].**

| Molecule | Position | System | $E_{ads}$ | $\Delta_Q$ | System | $E_{ads}$ | $\Delta_Q$ |
|---|---|---|---|---|---|---|---|
| SAM | Benzene | Pristine ZZ | -0.279 | 0.033 | SV ZZ | -0.509 | -0.023 |
| | | Pristine AM | -0.312 | 0.023 | SV AM | -0.373 | 0.002 |
| SDZ | Benzene | Pristine ZZ | -0.217 | 0.018 | SV ZZ | -0.382 | -0.006 |
| | | Pristine AM | -0.285 | -0.003 | SV AM | -0,537 | 0.017 |
| SMX | Benzene | Pristine ZZ | -0.207 | -0.001 | SV ZZ | -0.372 | 0.000 |
| | | Pristine AM | -0.248 | 0.002 | SV AM | -0.322 | -0.021 |

Benzene sites) BPNTs, both for pristine and SV defect-laden configurations. They are plotted in Fig 3. First, we comment on the band structures corresponding to clean nanotubes (see S2 Fig in Supporting information). In this case, our results are in accordance with previously reported ones [45,46]: Pristine (14, 14) AM BPNT is a direct gap semiconductor with $E_{gap}$ = 1.793 eV, whilst (14, 0) ZZ BPNT is an indirect gap semiconductor $E_{gap}$ = 1.927 eV. The presence of a SV significantly modifies this picture, with a magnetization of the system been induced. An overall reduction in the band gap value occurs, with the appearance of localized states within the gap region, thus transforming ZZ and AM BPNTs systems into small gap magnetic bipolar semiconductors. An additional particular feature is that defect-laden AM BPNT becomes, in this case, an indirect gap system.

Adsorption of antibiotics molecules in the pristine case manifests the appearance of dopant-like states above the valence band. In general, there is a slight reduction of the gap in all cases, keeping its direct or indirect nature, with the exception of SAM-adsorbed ZZ BPNT where a small increase of $E_{gap}$ to 2.0 eV is appreciated. Besides the presence of localized states (some of them almost dispersionless in the ZZ case), the main result on the electronic structure of the adsorption of considered sulfonamide molecules on SV defective BPNTs is that, when either a SAM or SMX molecule is adsorbed, AM BPNT shows a transition from a bipolar-magnetic semiconductor to non magnetic metallic character. This suggests that defective AM BPNTs also can be employed as a sensor for these antibiotics. Fig 4 shows the total density of states (DOS) for all considered complexes; these results are in agreement with the band structure and also suggest that in the pristine ZZ and AM BPNTs molecules transfer charge to the nanotube with the DOS been shifted. When a single vacancy is present, the magnetic states of the ZZ BPNTs are maintained. Still, for AM BPNTs, the magnetic state disappears, suggesting an effect of nanotube chirality on the magnetic properties induced by single vacancies. In addition, a Bader analysis was used to determine the charge transfer between the BPNTs and the sorbed sulfonamides. Positive values of $\Delta Q$ indicate that the molecule transfers charge to the BPNT, while negative values imply that the molecule receives charge from the BPNT. The results of $\Delta Q$ for the most probable adsorption configuration are present in Table 2; these results indicate that the transfer of charge depends on BPNT chirality numbers and the type of sulfonamide molecule involved.

## Optical response

To further analyze the properties of molecule adsorption onto the surface of BPNTs, we evaluate the optical response of the complex. In this sense, the imaginary part of the dielectric function ($\epsilon_2$), which is proportional to light absorption, is calculated within the independent particle approximation. In Fig 5, we show $\epsilon_2$ as a function of incident photon energy for SAM, SDZ, and SMX sulfonamides adsorbed on ZZ and AM pristine and defective BPNTs. In the

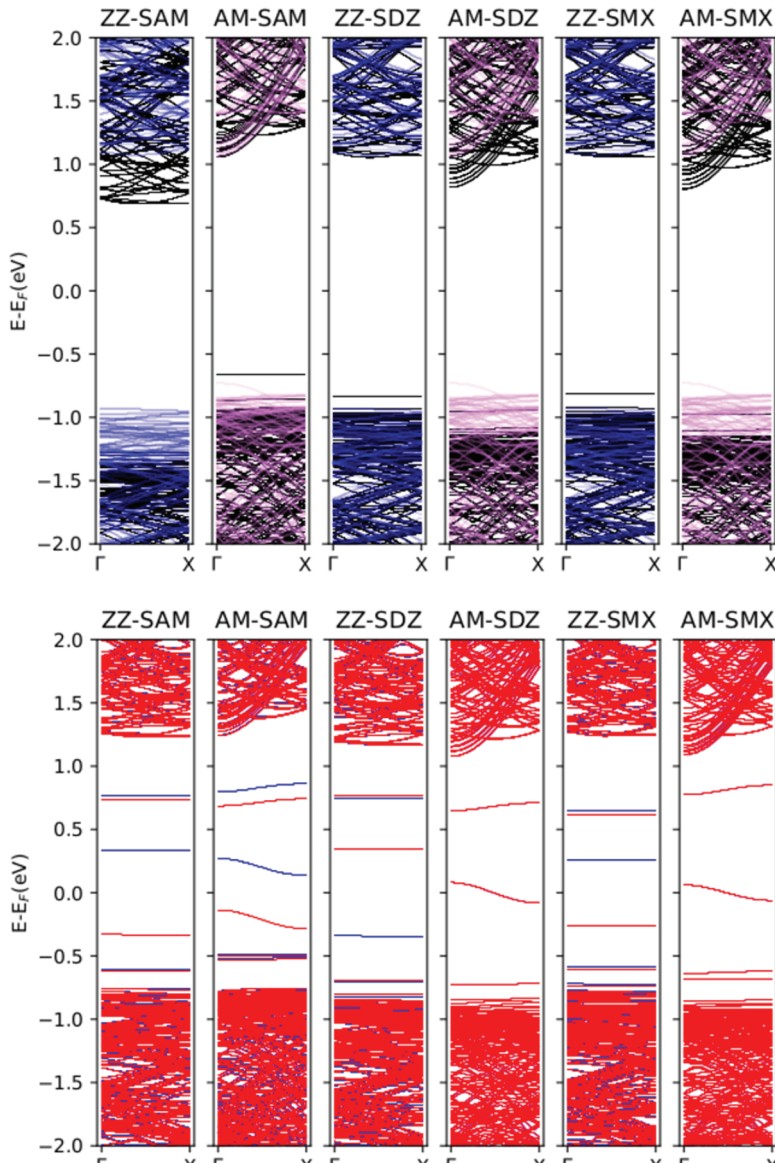

**Fig 3. Band structure of armchair and zigzag blue phosphorene nanotubes and the same for the complexes formed of a blue phosphorene nanotube plus adsorbed SAM, SDZ or SMX molecules at Benzene sites.** The band structure of pristine nanotubes is include in the top panel as translucent violet and blue lines.

https://doi.org/10.1371/journal.pcbi.0312034.g003

plots, each peak indicates a possible transition from one valence state to one conduction state. Peaks may come from the nanotube or the molecule and can even be derived from hybridized states. Such transition can be also dominated by the polarization of incident light. In this case, we observe a larger peak structures for the case of light polarized parallel to the BPNT axis, and when the molecule is adsorbed, the changes in the $\epsilon_2$ spectrum are better appreciated for pristine AM-BPNT. This is in agreement with the results of the electronic structure previously described, when the most significant changes are induced in the bands around $\Gamma$ point. However, when the BPNTs have structural defects as vacancies, these imperfections can shield

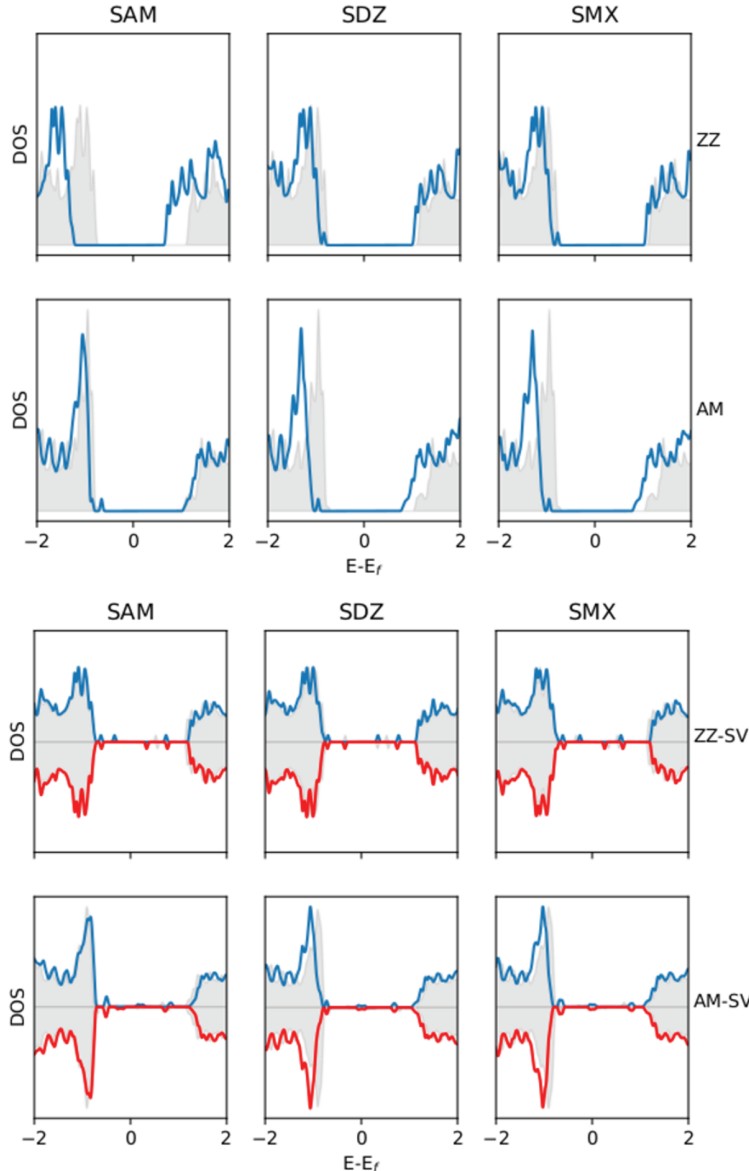

**Fig 4. Total density of sates of armchair and zigzag blue phosphorene nanotubes and the same for the complexes formed of a blue phosphorene nanotube plus adsorbed SAM, SDZ or SMX molecules at Benzene sites.** The gray shaded areas are the density of states of the single nanotubes.

https://doi.org/10.1371/journal.pcbi.0312034.g004

the effects of the interaction with the molecule, making it difficult to detect them in the optical spectrum. In general, when the electromagnetic signal is perpendicularly polarized there are no notably prominent features in the absorption spectra, perhaps with the exception of SAM-adsorbed ZZ pristine BPNTs.

## Recovery time

The adsorption energies determined for the interaction of antibiotics on BPNTs directly influence the recovery time expended by the BPNT in returning to its original state. From

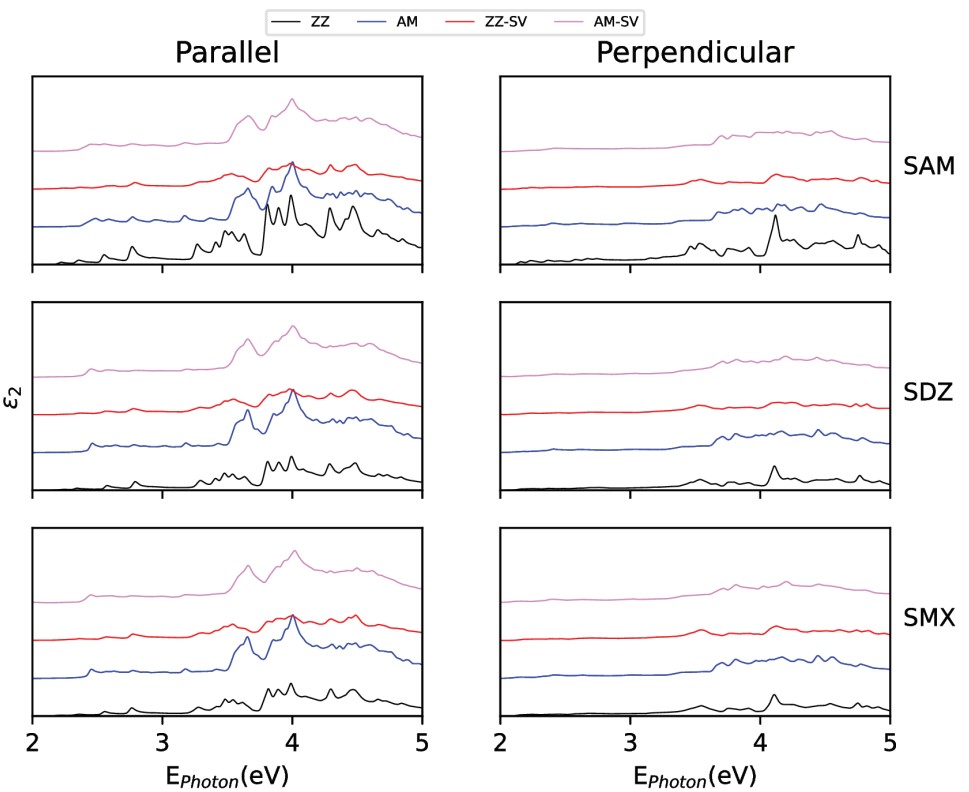

**Fig 5. Imaginary part of the dielectric function for SAM, SDZ and SMX molecules adsorbed onto zigzag and armchair pristine and defective blue-phosphorene nanotubes.** Two polarization of the insident light are considere, parallel and perpendicular to nanotube axis.

https://doi.org/10.1371/journal.pcbi.0312034.g005

Table 3, it can be seen than in all cases considered the initial form of the BPNTs, in both pristine and SV configurations, can be attained quickly. In particular, it would take little time for the BPNT to be retrieved back to its premier position. However, it is important to highlight that both the vary large and very small values of $\tau$ are unfavorable for detection in real experiments [30]. The values for other adsorption modes are in the Supporting information Table S3.

## Solvation effect

The adsorption energies for gas and water phases are considered for SAM, SDZ, and SMX molecules adsorbed on passivated fragments of ZZ and AM BPNTs. Obtained results appear in Table 4. As mentioned, calculations employed the empirical xTB approximation with the GFN1-xTB parameterization. Recently, a systematic study shows that GFN1-xTB under a generalized Born model with surface area (GBSA) is an excellent tool to investigate the solvation effect for molecules [37]. In general, our results suggest that water solvation reduces the adsorption energies for all molecules compared to the gas phase. Also, it is essential to highlight that the adsorption energies obtained with GFN1-xTB are overhead of the vdW-DFT. However, this appears as a result of the empirical parameterization and the finite cluster model employed. If we analyzed the percentage changes induced in the adsorption energy, we would observe a maximum percent variation close to 37%, with only the SDZ molecule on defective zigzag nanotube exceeding the 50% percent of reduction in the adsorption energy,

**Table 3. Results interaction of Benzene sites for pristine and single-vacancy (SV) with SAM, SDZ and SMX molecules; recovery time ($\tau$) is presented in $s$.**

| | | | $\theta_0 = 10^{12}$ (Visible light) | | | |
|---|---|---|---|---|---|---|
| Molecule | Position | System | $\tau$ 300K | $\tau$ 310K | $\tau$ 320K | $\tau$ 330K |
| SAM | Benzene | Pristine ZZ | $4,9 \times 10^{-08}$ | $3,4 \times 10^{-08}$ | $2,5 \times 10^{-08}$ | $1,8 \times 10^{-08}$ |
| | | Pristine AM | $1,8 \times 10^{-07}$ | $1,2 \times 10^{-07}$ | $8,3 \times 10^{-08}$ | $5,9 \times 10^{-08}$ |
| | | SV ZZ | $4,8 \times 10^{-04}$ | $2,5 \times 10^{-04}$ | $1,4 \times 10^{-04}$ | $7,8 \times 10^{-05}$ |
| | | SV AM | $1,9 \times 10^{-06}$ | $1,2 \times 10^{-06}$ | $7,6 \times 10^{-07}$ | $5,0 \times 10^{-07}$ |
| SDZ | Benzene | Pristine ZZ | $4,4 \times 10^{-09}$ | $3,3 \times 10^{-09}$ | $2,6 \times 10^{-09}$ | $2,0 \times 10^{-09}$ |
| | | Pristine AM | $6,0 \times 10^{-08}$ | $4,2 \times 10^{-08}$ | $3,0 \times 10^{-08}$ | $2,2 \times 10^{-08}$ |
| | | SV ZZ | $2,6 \times 10^{-06}$ | $1,6 \times 10^{-06}$ | $1,0 \times 10^{-06}$ | $6,8 \times 10^{-07}$ |
| | | SV AM | $1,1 \times 10^{-03}$ | $5,5 \times 10^{-04}$ | $2,9 \times 10^{-04}$ | $1,6 \times 10^{-04}$ |
| SMX | Benzene | Pristine ZZ | $3,0 \times 10^{-09}$ | $2,3 \times 10^{-09}$ | $1,8 \times 10^{-09}$ | $1,4 \times 10^{-09}$ |
| | | Pristine AM | $1,5 \times 10^{-08}$ | $1,1 \times 10^{-08}$ | $8,2 \times 10^{-09}$ | $6,2 \times 10^{-09}$ |
| | | SV ZZ | $1,8 \times 10^{-06}$ | $1,1 \times 10^{-06}$ | $7,3 \times 10^{-07}$ | $4,8 \times 10^{-07}$ |
| | | SV AM | $2,6 \times 10^{-07}$ | $1,7 \times 10^{-07}$ | $1,2 \times 10^{-07}$ | $8,2 \times 10^{-08}$ |
| | | | $\theta_0 = 10^{16}$ (UV light) | | | |
| Molecule | Position | System | $\tau$ 300K | $\tau$ 310K | $\tau$ 320K | $\tau$ 330K |
| SAM | Benzene | Pristine ZZ | $4,9 \times 10^{-12}$ | $3,4 \times 10^{-12}$ | $2,5 \times 10^{-12}$ | $1,8 \times 10^{-12}$ |
| | | Pristine AM | $1,8 \times 10^{-11}$ | $1,2 \times 10^{-11}$ | $8,3 \times 10^{-12}$ | $5,9 \times 10^{-12}$ |
| | | SV ZZ | $4,8 \times 10^{-08}$ | $2,5 \times 10^{-08}$ | $1,4 \times 10^{-08}$ | $7,8 \times 10^{-09}$ |
| | | SV AM | $1,9 \times 10^{-10}$ | $1,2 \times 10^{-10}$ | $7,6 \times 10^{-11}$ | $5,0 \times 10^{-11}$ |
| SDZ | Benzene | Pristine ZZ | $4,4 \times 10^{-13}$ | $3,3 \times 10^{-13}$ | $2,6 \times 10^{-13}$ | $2,0 \times 10^{-13}$ |
| | | Pristine AM | $6,0 \times 10^{-12}$ | $4,2 \times 10^{-12}$ | $3,0 \times 10^{-12}$ | $2,2 \times 10^{-12}$ |
| | | SV ZZ | $2,6 \times 10^{-10}$ | $1,6 \times 10^{-10}$ | $1,0 \times 10^{-10}$ | $6,8 \times 10^{-11}$ |
| | | SV AM | $1,1 \times 10^{-07}$ | $5,5 \times 10^{-08}$ | $2,9 \times 10^{-08}$ | $1,6 \times 10^{-08}$ |
| SMX | Benzene | Pristine ZZ | $3,0 \times 10^{-13}$ | $2,3 \times 10^{-13}$ | $1,8 \times 10^{-13}$ | $1,4 \times 10^{-13}$ |
| | | Pristine AM | $1,5 \times 10^{-12}$ | $1,1 \times 10^{-12}$ | $8,2 \times 10^{-13}$ | $6,2 \times 10^{-13}$ |
| | | SV ZZ | $2,6 \times 10^{-11}$ | $1,7 \times 10^{-11}$ | $1,2 \times 10^{-11}$ | $8,2 \times 10^{-12}$ |
| | | SV AM | $1,8 \times 10^{-10}$ | $1,1 \times 10^{-10}$ | $7,3 \times 10^{-11}$ | $4,8 \times 10^{-11}$ |

**Table 4. Calculated adsorption energies in gas phase ($E_{ads} - gas$) and water ($E_{ads} - sol$) for SAM, SDZ, and SMX molecules adsorbed on blue phosphorene nanotube fragment employing xTB. The Gibbs free solvation energy and the change of the adsorption energy ($\Delta E_{ads}$) are present. All results are in eV.**

| Molecule | Position | System $E_{ads} - gas$ | $E_{ads} - sol$ | $\Delta E_{ads}$ | $G_{sol}$ |
|---|---|---|---|---|---|
| SAM | Benzene | Pristine ZZ | −0.870 | −0.563 | 0.307 | −2.396 |
| | | Pristine AM | −1.226 | −0.801 | 0.425 | −2.657 |
| | | SV ZZ | −1.182 | −0.776 | 0.406 | −2.287 |
| | | SV AM | −1.069 | −0.907 | 0.162 | −2.665 |
| SDZ | Benzene | Pristine ZZ | −1.298 | −0.951 | 0.347 | −2.416 |
| | | Pristine AM | −1.189 | −0.870 | 0.319 | −2.885 |
| | | SV ZZ | −1.035 | −0.374 | 0.661 | −2.522 |
| | | SV AM | −1.538 | −1.205 | 0.333 | −2.837 |
| SMX | Benzene | Pristine ZZ | −1.369 | −1.086 | 0.283 | −2.563 |
| | | Pristine AM | −1.515 | −1.358 | 0.157 | −3.183 |
| | | SV ZZ | −1.547 | −1.181 | 0.366 | −2.429 |
| | | SV AM | −1.627 | −1.322 | 0.305 | −2.950 |

when the solvent is considered. In accordance, if we assume that the adsorption energies obtained with vdW-DFT also undergo a similar percent decreasing in the presence of water, this reduction would not affect the capability of the system to sensing sulfonamides.

## Conclusions

In this work, we have investigated the features of three different sulfonamide molecule adsorption onto armchair and zigzag blue-phosphorene nanotubes. To improve this process,

a *P* atom of the BPNT has been removed given a single-vacancy defect. It is observed that the presence of such a vacancy enhances the reactivity of the BPNTs so that the adsorption energies increased for all evaluated systems to about a percentage ranges that go from 20% to 89% for the AM SAM (smaller increase) and AM SDZ (larger increase) systems, respectively. This indicates that structural defects improve the capacity of BPNTs for adsorption of these toxic antibiotics. The orders of $E_{ads}$ of three sulfonamides were as follows: SAM > SDZ > SMX for pristine systems and ZZ SV. In SV AM-BPNTs SDZ molecule presents higher $E_{ads}$ than SAM and SMX (SDZ > SAM > SMX). The calculations results demonstrate that despite the increase in the reactivity of the ZZ SV BPNT to the sulfonamides, AM configurations show a transition from bipolar-magnetic semiconductor to not magnetic metallic system, suggesting that defective AM BPNTs also can be employed as a sensor for antibiotic molecules like SAM, SDZ, and SMX. The electronic responses of zigzag (ZZ) and armchair (AM) blue phosphorene nanotubes (BPNTs) to sulfonamide-based compounds were investigated using Density Functional Theory (DFT) calculations. The analysis focused on three molecules: SAM, SDZ, and SMX. The findings revealed that these molecules weakly adsorb onto pristine BPNTs, with adsorption energies of approximately -0.312, -0.285, and -0.248 eV, respectively. SAM exhibited the highest adsorption energy in all configurations considered, particularly at the Benzene position. Besides, studies of solvation indicate that, although adsorption is reduced by the presence of water, blue-phosphorene nanotubes remain as quite plausible means to remove toxic antibiotic molecules from environment.

## Supporting information

**S1 Text. Extra information with figures and tables.**
(PDF)

**S2 CIF Files. CIF data of the equilibrium structures.**
(ZIP)

## Acknowledgments

The authors would like to thank Universidad de Medellín (UdeM) for supporting their work.

## Author contributions

**Conceptualization:** Julian D. Correa, Miguel E. Mora-Ramos, Elizabeth Flórez.

**Formal analysis:** Jóse M. Vergara.

**Investigation:** Jóse M. Vergara.

**Methodology:** Julian D. Correa, Elizabeth Flórez.

**Software:** Jóse M. Vergara.

**Supervision:** Julian D. Correa, Elizabeth Flórez.

**Writing – original draft:** Jóse M. Vergara.

**Writing – review & editing:** Julian D. Correa, Miguel E. Mora-Ramos, Elizabeth Flórez.

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
