## [Decision Letter · Decision Letter 0]

31 Jul 2024

PONE-D-24-25556Unraveling the influence of defects on Sulfonamide adsorption onto Blue-phosphorene nanotubePLOS ONE

Dear Dr. Correa,

Thank you for submitting your manuscript to PLOS ONE. After careful consideration, we feel that it has merit but does not fully meet PLOS ONE’s publication criteria as it currently stands. Therefore, we invite you to submit a revised version of the manuscript that addresses the points raised during the review process.

We look forward to receiving your revised manuscript.

Kind regards,

Niravkumar Joshi

Academic Editor

PLOS ONE

Journal Requirements:

Reviewers' comments:

Reviewer's Responses to Questions

**Comments to the Author**

1. Is the manuscript technically sound, and do the data support the conclusions?

Reviewer #1: No

Reviewer #2: Yes

2. Has the statistical analysis been performed appropriately and rigorously? 

Reviewer #1: No

Reviewer #2: Yes

3. Have the authors made all data underlying the findings in their manuscript fully available?

Reviewer #1: Yes

Reviewer #2: Yes

4. Is the manuscript presented in an intelligible fashion and written in standard English?

Reviewer #1: No

Reviewer #2: Yes

5. Review Comments to the Author

Reviewer #1: The manuscript titled "Unraveling the Influence of Defects on Sulfonamide Adsorption onto Blue-Phosphorene Nanotube" requires significant revisions to meet the quality standards of PLOS ONE. Below are specific comments and suggestions for improvement:

1. The introduction should clearly highlight the novelty of the present work compared to existing literature. It is crucial to discuss previous studies in relation to the present work, preferably using a comparative table format.

2. The present work solely relies on theoretical calculation therefore, the title needs modification. Authors are advised to add “Density Functional Theory” in the title for clarity and to accurately reflect the scope of the research.

3. What is the role of nanostructured carbon materials (nanotubes) in comparison to other nanostructures?

4. Introduction need rigorous modification, several sentences are incomplete and contain grammatical mistakes (e.g., line no 35, 4).

5. What is role of zigzag and armchair configuration for BPNTs?

6. Authors are advised to provide details on the different adsorption sites on nanotubes for calculating the adsorption energy and Bader charge. Additionally, consider including density of state calculations to enhance the understanding of electronic structures in relation to adsorption mechanisms.

Reviewer #2: The author applied density functional theory to calculate the efficiency of (14,14) armchair and (14,0) zigzag BPNTs as adsorbents for three popular toxic antibiotics, SAM, SMX, and SDZ from water bodies. The manuscript is well written overall.

Here are some minor points that can be improved:

1. Lines 15-19: Avoid unclear descriptions such as "advantage of the potential of some structures" and "to remove organic contaminants from wastewater due to their properties." Please be specific about which structures and properties you are referring to.

2. Lines 29-32: What does the structure of the BP looks like? It would be better to include a picture of the four types of 2D allotropes of phosphorus. This is important for readers to understand the context.

3. Line 87: Note that 300 K is not typically considered ambient temperature.

4. Line 246: some typo here "SAM ¿ SDZ ¿ SMX".

5. Line 202: "salvation effect" Is that a typo?

6. Table 4: Supposed to be moved to under "salvation effect".

Sincerely,

6. PLOS authors have the option to publish the peer review history of their article (what does this mean?). If published, this will include your full peer review and any attached files.

Reviewer #1: No

Reviewer #2: No

---

## [Author Response · Author response to Decision Letter 1]

18 Sep 2024

Reviewer #1: The manuscript titled "Unraveling the Influence of Defects on Sulfonamide Adsorption onto Blue-Phosphorene Nanotube" requires significant revisions to meet the quality standards of PLOS ONE. Below are specific comments and suggestions for improvement:

1. The introduction should clearly highlight the novelty of the present work compared to existing literature. It is crucial to discuss previous studies in relation to the present work, preferably using a comparative table format.

The authors: We deeply thank the reviewer his/her comments and, in particular, for this valuable suggestion. In accordance, we have revised the introductory section by incorporating new references and providing a clearer discussion. The new leading paragraph in the introduction is given the following:

“Numerous methods are utilized to remove antibiotics from water, including biological processes, advanced oxidation processes (AOPs), and adsorption processes. Adsorption is considered one of the most advantageous techniques due to its simplicity, economy, and easy operation. For antibiotic remediation, a wide range of adsorbents have been used [2, 3]. Usually, they involve activated carbon and carbonaceous materials. Notably, carbon-based materials, such as biochar, graphene, and nanotube carbon have emerged as the most promising solutions for removing antibiotics from contaminated waters. Active carbon is characterized by its functionality, porosity, surface morphology, and chemistry, which enhance the adsorption process. [5]. On the other hand,two-dimensional carbon materials such as graphene or graphene oxides show an affinity for removing various classes of organic contaminants from water due to their highly hydrophobic surface, open-layer morphology, and high adsorption affinity [6]. It has been discovered that carbon nanotubes have the potential to be used in removing antibiotics from water. This is due to their cost-effectiveness, lower energy requirements, minimal chemical usage, and environmental impact, as well as their large surface area and increased chemical reactivity [7, 8]. The adsorption of Sulfanilamide (SAM), Sulfamerazine (SMR), Sulfadimethoxine (SMX), Sulfadiazine (SDZ), Sulfamethazine (SMT), and Sulfamethoxydiazine (SMD) on carbon nanotubes has been explored in several works [9–11]. In the case of SMX, it is shown that the π - π interaction was one of the mechanisms for SMX adsorption on multi-wall carbon nanotubes, and that, in general, the antibiotics are adsorbed in CMs through various non-covalent interactions, including van der Waals dispersion π - π interactions, hydrophobic interaction, and hydrogen bonding [9]. Theoretical calculations based on density functional theory show that SDZ and SMX sulfonamides are adsorbed on single wall carbon nanotube with adsorption energies of −0.566 eV and −0.551 eV, respectively, but when a water environment is considered these energies increase [11]. More recently, Liu et al. Have shown that carbon nanotubes have the potential to efficiently remove sulfonamides from aqueous solutions through the adsorption process, reaching a high efficiency (in a pH adsorption range of 3 to 9). These studies show that carbon nanotubes have a promising potential as an effective adsorbent for removing sulfonamide antibiotics from aqueous solutions. However, these processes require further investigation, with the inclusion of novel materials, to guide engineering applications since removing antibiotics can be, in some cases, incomplete [12, 13].”

2. The present work solely relies on theoretical calculation therefore, the title needs modification. Authors are advised to add “Density Functional Theory” in the title for clarity and to accurately reflect the scope of the research.

The authors: Following the indication, we have changed the manuscript’s title to

“Unraveling the influence of defects on Sulfonamide adsorption onto Blue-phosphorene nanotube using density functional theory”

3. What is the role of nanostructured carbon materials (nanotubes) in comparison to other nanostructures?

The authors: To properly comment on this subject, we have included a couple of sentences at the end of the newly written in the introduction.

4. Introduction need rigorous modification, several sentences are incomplete and contain grammatical mistakes (e.g., line no 35, 4).

The authors: We have carefully checked those mistakes and corrected the English writing in the section.

5. What is role of zigzag and armchair configuration for BPNTs?

The authors: Similar to graphene, blue phosphorene monolayers have two symmetric directions labeled as zigzag and armchair. When these monolayers are rolled to form a nanotube, if the roll is along the zigzag or armchair symmetric direction, the nanotubes are achiral. Working along these directions could modulate the electronic properties because, in each case, we encounter different geometric nanotube configurations. For reference, we respectfully suggest consulting the following references: 10.1016/j.cocom.2022.e00701, 10.1021/acs.jpcc.5b12112

 In addition, to properly clarify this aspect, we have included a new sentence in the computational section that reads

“Distinct chiral numbers defined different electronic properties of the BPNTs.”

6. Authors are advised to provide details on the different adsorption sites on nanotubes for calculating the adsorption energy and Bader charge. Additionally, consider including density of state calculations to enhance the understanding of electronic structures in relation to adsorption mechanisms.

The authors: Thanks again to the reviewer for this comment.

Regarding the various adsorption sites, we took into account the prevalence of the adsorption mode over the site in the nanotube because variations in the adsorption site lead to minor changes in the adsorption energy compared to the adsorption mode. For instance, when Sam is adsorbed on BPNT 14,0 at two different sites using xTB, we obtained the following results:

In relation with the total density of states, we have included a new Figure and updated the band structure plots. Also we added a new paragraph at the end of the discussion of Electronic Properties. It reads

Figure 4 shows the total density of states (DOS) for all complexes; these results are in agreement with the band structure and also suggest that in the pristine ZZ and AM BPNTs, the molecules transfer charge to nanotube due to that the DOS is leaf-shifted. Now, when a single vacancy is present, the magnetic states of the ZZ BPNTs are maintained. Still, for AM BPNTs, the magnetic state is lost, suggesting an effect of the chirality of the nanotubes on the magnetic properties induced for single vacancies.

Also, the calculated charge transfer, ∆Q, was include as a new column in the Table 2, and a subsequent paragraph on these results appears following the above presented one in the text. It reads

In addition, a Bader analysis was used to determine the charge transfer between the BPNTs and the sorbed sulfonamides. Positive values of ∆Q indicate that the molecule transfers charge to the BPNT, while negative values imply that the molecule receives charge from the BPNT. The results of ∆Q for the most probable adsorption configuration are present in Tab. 1; these results indicate that the transfer of charge depends on the BPNT chirality numbers and the type of sulfonamide molecule involved.

Reviewer #2: The author applied density functional theory to calculate the efficiency of (14,14) armchair and (14,0) zigzag BPNTs as adsorbents for three popular toxic antibiotics, SAM, SMX, and SDZ from water bodies. The manuscript is well written overall.

Here are some minor points that can be improved:

1. Lines 15-19: Avoid unclear descriptions such as "advantage of the potential of some structures" and "to remove organic contaminants from wastewater due to their properties." Please be specific about which structures and properties you are referring to.

The authors: We deeply thank the reviewer for his/her comments to our work. We have revised the introduction to ensure clarity.

2. Lines 29-32: What does the structure of the BP looks like? It would be better to include a picture of the four types of 2D allotropes of phosphorus. This is important for readers to understand the context.

The authors: In regard to this question, a new figure has been added to the supplementary material along with a comment inserted into the text.

3. Line 87: Note that 300 K is not typically considered ambient temperature.

The authors: All typos have been corrected.

4. Line 246: some typo here "SAM ¿ SDZ ¿ SMX".

The authors: All typos have been corrected.

5. Line 202: "salvation effect" Is that a typo?

The authors: All typos have been corrected.

6. Table 4: Supposed to be moved to under "solvation effect".

The authors: Done.

---

## [Decision Letter · Decision Letter 1]

30 Sep 2024

Unraveling the influence of defects on Sulfonamide adsorption onto Blue-phosphorene nanotube using density functional  theory

PONE-D-24-25556R1

Dear Dr. Julian Correa,

We’re pleased to inform you that your manuscript has been judged scientifically suitable for publication and will be formally accepted for publication once it meets all outstanding technical requirements.

Kind regards,

Niravkumar Joshi

Academic Editor

PLOS ONE

Additional Editor Comments (optional):

Reviewers' comments:

Reviewer's Responses to Questions

**Comments to the Author**

1. If the authors have adequately addressed your comments raised in a previous round of review and you feel that this manuscript is now acceptable for publication, you may indicate that here to bypass the “Comments to the Author” section, enter your conflict of interest statement in the “Confidential to Editor” section, and submit your "Accept" recommendation.

Reviewer #1: (No Response)

Reviewer #2: All comments have been addressed

2. Is the manuscript technically sound, and do the data support the conclusions?

Reviewer #1: (No Response)

Reviewer #2: Yes

3. Has the statistical analysis been performed appropriately and rigorously? 

Reviewer #1: (No Response)

Reviewer #2: Yes

4. Have the authors made all data underlying the findings in their manuscript fully available?

Reviewer #1: (No Response)

Reviewer #2: Yes

5. Is the manuscript presented in an intelligible fashion and written in standard English?

Reviewer #1: (No Response)

Reviewer #2: Yes

6. Review Comments to the Author

Reviewer #1: (No Response)

Reviewer #2: All issues mentioned in the first review have been addressed. Please check typos and reformat to fit the journal requirements before the final publication.

Thanks,

7. PLOS authors have the option to publish the peer review history of their article (what does this mean?). If published, this will include your full peer review and any attached files.

Reviewer #1: No

Reviewer #2: No

---

## [Editor Report · Acceptance letter]

PONE-D-24-25556R1

PLOS ONE

Dear Dr. Correa,

I'm pleased to inform you that your manuscript has been deemed suitable for publication in PLOS ONE. Congratulations! Your manuscript is now being handed over to our production team.

Kind regards,

on behalf of

Dr. Niravkumar Joshi

Academic Editor

PLOS ONE